# Effect of Radial Moisture Distribution on Frequency Domain Dielectric Response of Oil-Polymer Insulation Bushing

**DOI:** 10.3390/polym12061219

**Published:** 2020-05-27

**Authors:** Daning Zhang, Guanwei Long, Yang Li, Haibao Mu, Guanjun Zhang

**Affiliations:** State Key Laboratory of Electrical Insulation and Power Equipment, Xi’an Jiaotong University, Xi’an 710049, China; zdn_sdu@163.com (D.Z.); xjlgw6@stu.xjtu.edu.cn (G.L.); ly416159091@stu.xjtu.edu.cn (Y.L.)

**Keywords:** oil-immersed polymer, moisture content, interface polarization, space charge polarization

## Abstract

In order to realize the diagnosis of water distribution, this paper analyzes the interface polarization and macroscopic space charge polarization mechanism when the water distribution is non-uniform. The experimental results of this paper and bushing show that when the moisture distribution is non-uniform, there is a significant loss peak in the tan*δ*-*f* curve. The loss peak shifts to higher frequencies as the non-uniformity coefficient increases. There are common intersection points between multiple tan*δ*-*f* curves. Further, this paper realizes the diagnosis of the location of moisture distribution through Frequency Domain Spectroscopy (FDS) testing of different voltages and different wiring methods based on the macroscopic space charge polarization. In the single-cycle FDS test, when the positive electrode is first added to the area with higher moisture content, the amplitude of the tan*δ*-*f* curve is smaller. The tan*δ*-*f* curves under different wiring methods constitute a “ring-shaped” loss peak. As the voltage increases, the peak value of the loss peak shifts to the lower frequency band. As the temperature increases, the peak value of the loss peak shifts to higher frequencies. Based on the above rules and mechanism analysis, this research provides a new solution for the evaluation of moisture content of oil-immersed polymers equipment.

## 1. Introduction

As an important part of the external insulation of transformers, high-voltage oil-immersed polymer insulation bushings are characterized by large quantities, high prices, and excellent insulation properties. Therefore, the scientific and effective diagnostic evaluation of its insulation status determines the safety and stability of the high-voltage power grid. Capacitive bushings are composite insulation composed of insulating oil and polymer (cellulose) [1]. Moisture of the bushing is one of the main causes of its insulation failure. The accumulation of moisture in the insulation polymer significantly increases the partial discharge; and the breakdown field strength is greatly reduced, which in turn causes electrical accidents [2]. Due to the capacitive plate structure of the bushing, external water intrusion and incomplete initial drying will lead to non-uniform moisture distribution in the bushing. Local high moisture content (mc) can also lead to accelerated thermal aging of the polymer [3]. Therefore, it is necessary to diagnose the local moisture content of the bushing. Further, clarifying the relative position of high moisture in the capacitive screen will improve product design.

In recent years, a method for evaluating the insulation state of oil-immersed polymers based on dielectric response has been popularized, such as frequency domain spectroscopy (FDS) and polarization and depolarization current (PDC). These methods judge the aging and dampness of the oil-polymer insulation by detecting the polarization loss and conductivity loss of the dielectric under the electric field [4,5]. Some scholars have established a distributed parameter model to explain the dielectric response of cellulose insulation of transformers when thermal and moisture change dynamically [5]. However, the determination of the distribution parameters will be affected by the main insulation structure of the transformer. Therefore, reference [6] proposes a non-uniform aging evaluation method based on the modified Debye model. This method can be independent of the size and structure of the insulation. In reference [7], the influence of voltage on a FDS curve is found for the first time. It is pointed out that the Garton effect leads to the nonlinear phenomena under different voltages. Mac Donald pointed out that the physical mechanism of the Garton effect belongs to blocking electrode polarization in micro-pores [8]. Due to the small pore size, the relaxation time of space charge polarization is difficult to measure. Frood used the theory of space charge polarization to explain the experimental phenomena in deionized water [9]. In conclusion, the polarization caused by the ion aggregation of water in the oil-polymer under the electric field is expected to be used to realize the water localization. Besides, partial discharge can be used to locate the concentrated defects in the transformer, but this method is difficult to achieve quantitative evaluation for bushing [10].

This paper proposes a method of radial dampness assessment based on interface polarization and macroscopic space charge polarization in FDS. Combining theoretical analysis and the actual insulation deterioration of transformer bushings in the field, oil-polymer insulation samples of different moisture levels and types were prepared in this paper. With reference to high-voltage transformer bushings, a scaled-down model of bushing was prepared. A vacuum temperature control test platform was constructed. First, FDS tests were performed on oil-immersed polymer samples with different types of moisture distribution, and the curve characteristics and internal mechanism of uneven moisture distribution were clarified. Based on the above, FDS tests of bushing models under different voltage amplitudes, voltage polarities (wiring method), and temperatures were performed. The macroscopic space charge polarization caused by hydrolyzed ions when the moisture distribution is non-uniform is analyzed. The time constant reflecting the polarization of the macroscopic space charge was extracted by the difference curve of tan*δ*-*f* under different voltages and wiring method, and the inherent relationship between the time constant and the type of damp was established.

## 2. Oil-Polymer Sample and Bushing Model

### 2.1. Oil-Polymer Samples

According to the design principle of the bushing structure, the thickness of a single insulation layer of a general high-voltage capacitive bushing is about (1~1.2 mm), and the number of bushing insulation layers of 110~330 kV voltage level is 30~90 layers. The larger the number of insulating laminates of a capacitive bushing, the easier it is to control the distribution of the electric field. Based on the above basic principles, an insulation laminates with a thickness of 1.04 mm, a length of 600 mm, and a width of 100 mm was prepared in the laboratory, as shown in Figure 1. A single insulation laminate consists of 8 sheets of cellulose paper. According to the Oommen moisture balance curve, different moisture was obtained by controlling the drying time [11]. The cellulose polymer was fully impregnated with 45# Karamay mineral insulation oil with moisture content of 11 ppm for 48 h and then subjected to a dielectric test. Based on the IEC 60814 standard, the moisture contents of insulation laminate with different dry time were tested by the Karl Fischer coulometer KFT831 (Metrohm AG Ltd., Herisau, Switzerland). Single laminated oil-immersed polymer samples with different moisture contents (<0.41%, 1.1%, 2.03%, 2.84%, 3.91%, 5.08%, and 6.11%) were prepared.

### 2.2. Bushing Model

High-voltage bushings are usually constructed of multiple oil-immersed polymer laminates. There are aluminum foil electrodes between the laminates to make the electric field uniform. The 26 kV bushing model is a capacitor core model with a transparent sheath, as shown in Figure 2a. The capacitor core insulation structure includes four built-in uniform aluminum foil plates. The zero screen radius and zero screen plate length are 16.5 mm and 219 mm, respectively. In order to achieve the equivalent electric field strength of the model and the actual electric field strength of the 500 kV bushing, the thickness between the layers of the bushing and the length of the step are adjusted to meet the requirements, as shown in Table 1. The maximum radial field strength of the casing is 4.5 kV/mm. The maximum axial field strength at the upper end is 0.1 kV/mm, and the maximum axial field strength at the lower end is 0.43 kV/mm. The maximum deviation of the field strength is within 5%.

Similarly, bushing models with different moisture content distributions were obtained by controlling the drying time of the inner and outer oil-immersed polymers, as shown in Table 2. The moisture distribution diagram of 1# bushing and the pictures of 4 bushings are shown in Figure 2.

The dielectric response tester (by means of IDAX + VAX, MEGGER Co., Ltd., Stockholm, Sweden) is selected as the FDS test equipment during the experiment. The test frequency band is 1 m~5 k Hz, and the voltage range is 50~800 V. The experimental flow is shown in Figure 3.

## 3. Analysis of Measurement Results

This section is divided into two parts of the sample and model experiments. For the oil-polymer samples, it mainly includes dielectric characteristics of the combination of uniform and non-uniform moisture. The experiment of the bushing model mainly includes the positioning research when the moisture distribution is non-uniform.

### 3.1. Oil-Polymer Samples

#### 3.1.1. Uniform Moisture

The unit laminated oil-polymer samples with different moisture contents were prepared according to the method in Section 2. The FDS test is conducted using a frequency band ranging from 1 mHz to 5 kHz at 40 °C. After the FDS test, the moisture content of the paper was re-calibrated with the Metrohm KFT831 Coulometer. The FDS characteristic curve of the uniformly dampened oil-impregnated polymer sample is shown in Figure 4 and Figure 5. It can be seen from Figure 4 that with the increase of moisture content, the tan*δ*-*f* curves show an increasing trend as a whole, and the increase in the middle frequency band is significantly larger. For the oil-polymer samples with moisture content of 0.41%, 1.10%, 2.03%, and 2.84%, the overall curve of tan*δ*-*f* is in the shape of “checkmark.” For the samples with moisture content of 3.91%, 5.08%, and 6.11%, the increase in tan*δ*-*f* curve in the low frequency band is small, but the increase in the middle and high frequency band is obvious.

It can be seen from Figure 5a that as the moisture content increases, the *C*′-*f* curve remains unchanged in the high frequency band (100 Hz to 5 kHz), and the *C*′-*f* curve increases in the low frequency band (1 mHz to 100 Hz). It can be seen from Figure 5b that as the moisture content increases, the imaginary part *C*″-*f* curve of the complex capacitor is similar to the tan*δ*-*f* curve, and the overall trend is increasing. The *C*″-*f* curves with 0.41%, 1.10%, 2.03% and 2.84% presents a “checkmark” shape. The *C*″-*f* curves of the samples with the moisture content of 3.91%, 5.08%, and 6.11% are all “straight,” and the slope is approximately –1 in the low frequency band (1 mHz~10 Hz). The slope of 2.03% and 2.84% of oil impregnated paper *C*″-*f* curve in the low frequency band (1 mHz~1 Hz) is approximately –1. The imaginary part *C*″-*f* of the complex capacitor mainly characterizes the conduction loss and polarization loss process of oil-paper insulation. The conduction loss is mainly reflected in the low frequency band. According to theoretical derivation [12], when the frequency is low enough, the polarization process has sufficient time to complete. At this time, the conductivity process in the oil-paper insulation medium is dominant, and the corresponding low-frequency *C*″-*f* curve is a straight line with a slope of –1.

#### 3.1.2. Non-Uniform Moisture

The unit laminated oil-polymer sample is disassembled and reassembled into a unit laminate with an average moisture content of 0.75% to 4%. The thickness of the double-laminates sample with non-uniform moisture is consistent with that in Section 3.1.1. The specific combination is shown in Table 3. The non-uniformity coefficient *n* is defined as the degree of deviation of the local moisture content and the average moisture content as follows:(1)$$n=\frac{MC_{\max}-MC_{\min}}{MC_{avg}}$$
where *MC*_max_ is maximum moisture content of the oil-polymer unit laminate; *MC*_min_ moisture content of the oil-polymer unit laminate; and *MC*_avg_ is the average moisture content of oil-polymer unit laminate. The FDS test results are shown in Figure 6, Figure 7 and Figure 8, and the test temperature is 40 °C.

It can be obtained from Figure 6, Figure 7, and Figure 8a that the larger non-uniformity coefficient *n* of the sample combination, the greater the tan*δ*-*f* curve fluctuates around the uniform combination. Compared with the sample with uniform moisture, the dielectric loss of the tan*δ*-*f* curve of the non-uniform sample increases in the middle and high frequency bands, and decreases in the low frequency band. Compared with the sample with uniform moisture, there are obvious dielectric loss peaks for the two types of non-uniform moisture. In Figure 6a, the loss peak of the sample combination of 1.10% + 2.84% is between 0.01 Hz and 0.1 Hz, and the trend of the high frequency band curve is similar to the curve of 2.84%. The loss peak of the 0.41 + 3.91% sample combination is between 0.01 Hz and 0.1 Hz, which is significantly convex compared to the 2.84% curve. The tan*δ*-*f* curve with large non-uniformity coefficient *n* also has a large starting frequency of loss peak. The starting frequency corresponds to the interface polarization time constant of the sample combination. There is a common curve intersection point between the two types of non-uniform sample combination and the uniform sample. In Figure 6a, the frequency corresponding to the intersection point is approximately 0.0046 Hz. In Figure 7a, the frequency corresponding to the intersection point is approximately 0.15 Hz. In Figure 8a, the frequency corresponding to the intersection point is approximately 1 Hz.

From Figure 6, Figure 7, and Figure 8b, it can be seen that the larger the non-uniformity coefficient *n* of the sample combination, the more obvious the increase in the low frequency band of the real part of the complex capacitor *C*′-*f* curve, which corresponds to the larger the additional capacitance of the interface polarization. The *C*′-*f* curves of the real part of the complex capacitor form a “spindle”-shaped closed-loop curve at the low-frequency part. The frequency at the maximum diameter of the “spindle”-shaped curve corresponds to the frequency of the intersection of the curves in Figure 6, Figure 7, and Figure 8a. The imaginary part *C*′′-*f* curve of the complex capacitor is consistent with the trend of tan*δ*-*f* curves.

### 3.2. Bushing Model

#### 3.2.1. Non-Central Symmetric Distribution of Moisture

The dielectric response tester with the amplifier is used to perform the FDS test on the bushing model. The FDS test frequency range is 1 m Hz~5 k Hz; test temperature is 40 °C; peak test voltage range is 50 V~800 V. The test process only analyzes the voltage and current waveforms within one period when the excitation source frequency is lower than 1 Hz. When the AC voltage is removed, the polar molecules in the oil-paper insulation enter the depolarization process. Therefore, the depolarization current may affect subsequent measurements. Therefore, to avoid the impact between multiple tests, the interval between multiple tests is 10 min. Two wiring methods are used for comparative testing. The first wiring method is that the high voltage end of the FDS tester is connected to the terminal lamination lead of the bushing model, and the measuring end is connected to the conductive rod of the bushing model. The second connection method is that the high voltage end of the tester is connected to the conductive rod of the bushing model, and the measuring end is connected to the terminal lamination lead of the bushing model.

The test results for the four bushing models are shown in Figure 9. As can be seen from Figure 9, the two wiring methods have significantly different effects on the test results of the 1 # and 2 # models. In Figure 9a,b, the two wiring methods have no effect on the middle and high frequency bands of the curve, and the curves basically overlap. In the low frequency band (0.001 Hz~0.1 Hz), the two wiring methods are significantly different. For the 1 # model, the amplitude of the tan*δ*-*f* curve in wiring mode 1 is greater than the amplitude in wiring mode 2. The amplitude of the tan*δ*-*f* curve is smaller when the positive half cycle of the sinusoidal voltage is first applied to the area with higher moisture content. For the model 2 #, in Figure 9b, the amplitude of the tan*δ*-*f* curve of wiring method 2 is greater than the amplitude of wiring method 1. When the positive half cycle of the voltage is first added to the area with higher water content, the amplitude of tan*δ*-*f* curve is small. In Figure 9c,d, the two wiring methods have no effect on the full frequency band of the curve, and the two curves basically overlap. When the radial moisture content distribution is center-symmetric, changing the wiring method has no effect on the two curves.

#### 3.2.2. The Influence of Voltage

In order to further explore the reasons for the difference in tan*δ*-*f* curves caused by the two wiring methods, FDS test under different voltages was carried out for model 1 #. By changing the order of the field strength and the polarity of the electric field, the mechanism of the difference in tan*δ*-*f* curves is analyzed. The FDS test temperature is 50 °C. The test voltage range is: 50~800 V. The test results are shown in Figure 10.

It can be seen from Figure 10 that at the same voltage, the loss of connection mode 2 is less than that of connection mode 1. As the excitation voltage increases, the tan*δ*-*f* curve shows a decreasing trend in the low frequency band. As the excitation voltage increases, the difference in tan*δ*-*f* curve becomes more obvious when the wiring method is changed. It can also be drawn from the enlarged diagrams in Figure 10 that the difference between the tan*δ*-*f* curves of the two wiring methods increases first and then decreases as the frequency increases, thereby forming a “ring”. The tan*δ*-*f* curve increases first and then decreases with increasing frequency.

In order to highlight the influence rule of voltage, the tan*δ*-*f* curves of the two connection modes under the same excitation voltage in Figure 10 are made different, and Figure 11 is obtained. As the voltage increases, the overall amplitude of the tan*δ*-*f* curve difference increases and then decreases. The reason for the amplitude difference is as described above. It can be seen from Figure 11 that the difference between the two wiring methods has a loss difference peak, and the frequency of the difference peak moves to a lower frequency as the field strength increases.

#### 3.1.3. The Influence of Temperature

In this section, the influence of different wiring methods at different temperatures on the uneven radial damped bushing model is developed. The test results at 40 and 30 °C are shown in Figure 12a and Figure 13. The difference curve in Figure 12a is shown in Figure 12b.

It can be obtained from Figure 12 that the tan*δ*-*f* curve at 40 °C has a similar law with the curve in Figure 10. But as the temperature decreases, the frequency of loss peaks also decreases. Similarly, the difference between the two sets of curves under the same voltage in Figure 12a is obtained as Figure 12b. The main peak frequency of the loss peak in Figure 12b gradually decreases with increasing voltage, and the variation range is between 0.01 Hz and 0.03 Hz. When the test temperature is further reduced to 30 °C, the tan*δ*-*f* curve in Figure 13 only shows a significant difference when the frequency is less than 0.1 Hz. As the test temperature decreases, the frequency band of the curve difference decreases. The difference is that when test temperature is 30 °C, no obvious main peak of the loss peak can be observed in the tan*δ*-*f* curve.

Compare the tan*δ*-*f* curves of 800 V voltage at three different temperatures, as shown in Figure 14. When the test temperature is 50 °C, the tan*δ*-*f* curve of the two connection methods shows obvious loss peaks, and the difference caused by the two wiring method is mainly concentrated near the loss peak. It is assumed that the peak frequency of the tan*δ*-*f* curve difference at 30 °C is less than the test results of the other two sets of higher temperatures. As the temperature increases, the peak frequency at different temperatures moves toward the high frequency.

## 4. Discussion

### 4.1. Non-Uniform Moisture Distribution

When the moisture content of oil-polymer sample is large, the increase of water makes the number of ions dissociated by water molecules increase, and at the same time, water as an ideal solvent for weak electrolytes makes the degree of dissociation of impurity molecules increase. As a result, the conductivity loss increases significantly. In addition, water molecules, as a kind of dipole with strong polarity, produce polarization loss under the alternating electric field. In addition to the above, multilayer oil-paper insulation also has interfacial polarization losses caused by periodic changes in space charge between paper-paper composite interfaces with different water contents [6,13].

For a double-layer oil-impregnated paper insulation system, the thickness of each layer is *d*_1_, *d*_2_; the conductivity is *γ*_1_, *γ*_2_; the relative dielectric constant is *ε*_1_, *ε*_2_; and the cross-sectional area of the medium is *S*. The equivalent circuit of the oil-paper insulation system is shown in Figure 15 below.

If the externally applied voltage *U* is a DC voltage, when *t* = 0, the instantaneous distribution of the voltage depends on the capacitances *C*_1_ and *C*_2_. When *t* → ꝏ, the interface polarization process tends to a steady state. According to the law of current continuity, the voltage on the two layers of oil-paper insulation is distributed in proportion to *R*_1_ and *R*_2_. When an AC excitation voltage is applied to the double-layer oil-polymer sample, according to the superposition theorem, the full current under AC voltage excitation can be obtained as:(2)$$\dot{I}={\dot{I}}_{P}+{\dot{I}}_{C}=(\frac{1}{R}+\frac{\omega^{2}\theta^{2}g}{1+\omega^{2}\theta^{2}}+\frac{i\omega \theta g}{1+\omega^{2}\theta^{2}}+\frac{i\omega C_{1}C_{2}}{C_{1}+C_{2}})\dot{U}$$
where, ${\dot{I}}_{P}$ is the conduction loss current, and ${\dot{I}}_{C}$ is the capacitor current.
(3)$$\theta =\frac{R_{1}R_{2}(C_{2}+C_{1})}{R_{1}+R_{2}}$$
(4)$$g=\frac{{\left(R_{2}C_{2}-R_{1}C_{1}\right)}^{2}}{R_{1}R_{2}R{\left(C_{1}+C_{2}\right)}^{2}}$$
(5)$$R=R_{1}+R_{2}$$

The tangent of the dielectric loss angle is:(6)$$\tan\delta =\frac{I_{P}}{I_{c}}=\frac{\frac{1}{R}+\frac{\omega^{2}\theta^{2}g}{1+\omega^{2}\theta^{2}}}{\omega (C_{\infty }+\frac{\theta g}{1+\omega^{2}\theta^{2}})}$$

If we ignore the conductance loss of penetration in Equation (6), we can get the change trend of *C* and tan*δ* with frequency. When $\omega =1/\theta \sqrt{1+\theta g/C_{\infty }}$, the tan*δ*-ω curve has an extreme value as shown in Figure 16. This is also the reason for the spindle-shaped closed loop in the real part of the complex capacitor. On the other hand, the frequency at which the maximum value of tan*δ*-*f* also depends on *g*, *θ*, and the degree of non-uniform.

The X-Y model and polymer database are often used for moisture content assessment. It is also expected to assess the non-uniformity coefficient of moisture by introducing an FDS database of uneven moisture. With the introduction of multiple dimensions of data, there may be multiple optimal solutions for the evaluation results. By introducing the intersection of curves with the same average water content, the limitation of iterative evaluation can be increased.

The *C**-*f* curve database with approximate moisture content of 0% and variation gradient of moisture content of 0.1% was further obtained by interpolating the data of *C**-*f* curve of uniformly damp samples. According to the Equation (6) of interface polarization, the change rule of intersection frequency of curve under different average water content is further obtained, as shown in Figure 17. The changing law of intersection frequency of curve under different average water content is fitted, and the relationship between average water content and intersection frequency satisfies the following equation:(7)$$f_{c}=e^{\left(-17.20519+7.51008MC_{\mathrm{avg}}-0.80219MC_{\mathrm{avg}}{}^{2}\right)}$$
where, *f*_c_ is the frequency of the intersection point; *MC*_avg_ is the intersection frequency of tan*δ*-*f* curve with the same average moisture content.

### 4.2. Non-Central Symmetric Distribution of Moisture

In addition to the interface polarization between the two layers, there is also macroscopic space charge polarization caused by the accumulation of ion concentration. When an electric field is applied, free ions move directionally between the electrodes under the action of the electric field, thereby forming a conductive current. To make a non-zero current through the electrode, the electrode potential must deviate from the value of the balanced electrode potential. This phenomenon is called electrode polarization [14]. The ions accumulated near the electrode form a macroscopic space charge.

Taking the presence of more water H_2_O in paper as an example, the hydrolyzed ion H^+^ has a significantly different migration mechanism from other ions, so that the mobility of H^+^ ions differs greatly from other ions. According to the measurement data in the electrochemical method, the mobility of H^+^ ions in water at 25 °C is about 5 to 8 times greater than that of other ions [15]. In the liquid state, H^+^ ions exist in the form of H_3_O^+^. H_3_O^+^ can not only participate in the conductance process through the common ion movement method, but also form directional migration through proton exchange [15]. According to Grotthuss theory [16], H_3_O^+^ forms a new H_3_O^+^ by transferring protons to the adjacent water molecules, and the adjacent water molecules are hydrogen bonded. The schematic diagram of the proton transfer process is shown in Figure 18. From the above, for the solid-liquid composite medium with cellulose, the advantages of H^+^ ion proton conduction will be more obvious because the movement of ions is bound by cellulose based on Garton theory [17]. Correspondingly, the difference in mobility of positive and negative ions is more obvious.

For the ions dissociated by the weak electrolyte in the medium, the ions accelerate to the positive or negative electrode under the action of the electric field. Their direction of motion is the vector sum of the speed under the electric field plus the initial speed. The moving ions will last for a certain time before they collide with other carriers. This duration is called the relaxation time *τ_i_* of ionic conductivity [9].
(8)$$\tau_{i}=\frac{\varepsilon_{r}}{2q(n_{+}+n_{-})\mu_{\pm }}$$
where, *q* is the ionic charge; *n*_±_ is the number of ions; *μ*_±_ is the ion mobility; and *ε_r_* is the relative dielectric constant.

According to Equation (8), the relaxation time of ion conductivity is negatively correlated with ion concentration and ion mobility. According to Coelho’s theory of electrode polarization macroscopic space charge, the dielectric constant expression of space charge can be obtained [18].
(9)$$\varepsilon (\omega )=\varepsilon_{r}\frac{1+i\omega \tau_{i}}{i\omega \tau +\frac{1}{\eta_{w}}\tanh\eta_{w}}$$
where, $\eta_{w}=\eta_{0}\sqrt{1+iw\tau }$; $\eta_{0}=l/\lambda_{D}$; *μ*_±_ is the ion mobility, where *λ*_D_ is the Debye length of the ion, and when wτ≪1, $\eta_{w}\approx \eta_{0}$ Equation (9) can be transformed into
(10)$$\varepsilon (\omega )=\varepsilon_{\infty }+\frac{\varepsilon_{s}-\varepsilon_{\infty }}{1+i\omega \tau_{p}}$$

Equation (10) conforms to the Debye equation. From Equation (10), the relaxation time constant *τ*_p_ of space charge under electrode polarization can be obtained:(11)$$\tau_{p}=\eta_{0}\tau_{i}=\frac{d}{\mu_{\pm }}\sqrt{\frac{\varepsilon_{r}}{(n_{+}+n_{-})k_{b}T}}$$

According to Equation (11), the space charge relaxation time caused by ion accumulation is closely related to ion concentration, ion mobility, and temperature.

In Figure 12, the test results at different voltages with the same wiring method are caused by the blocking effect of cellulose on ion movement [19]. With different wiring methods, the difference of tan*δ*-*f* is mainly caused by the macroscopic space charge polarization. Figure 19 is a schematic diagram of the ion distribution of the double-layer damped non-uniform bushing model. Before the electric field is applied, the ion concentration in the paper layer with a larger moisture content is greater than that with a small moisture content. After the electric field is applied, when the positive electrode plate is close to the paper layer with a large moisture content, a large amount of negative ions accumulates near the positive electrode area; a small amount of positive ions accumulates near the negative electrode area.

After changing the polarity of the electric field, a large amount of positive ions accumulated near the negative electrode area; a small amount of negative ions accumulated near the positive electrode area. Since the mobility of positive ions (dominated by H^+^) is large, the positive ions accumulated at the negative electrode should be larger than the negative ions accumulated in the paper layer with large moisture (near the positive electrode) before the polarity reversal of the electric field. Since the accumulation and dissipation of ions are hysteresis relative to the polarity reversal of the applied electric field, the ion concentration in the positive and negative half cycles of the voltage is different under a complete cycle of the AC electric field. When the moisture distribution is non-uniform, the accumulation of ion concentration is more obvious, and the relaxation time constant is smaller.

### 4.3. Assessment Method

By extracting the characteristic values of the tan*δ*-*f* curves under different wiring methods and different test voltages, it can be further used to evaluate the radial distribution of moisture content of the oil-immersed polymer. As shown in Figure 20, the evaluation process first evaluates the non-uniformity coefficient and average moisture content by database comparison, and then conducts the identification and evaluation of the radial moisture distribution after determining that the bushing has non-uniform moisture distribution. The comparison database incorporates the XY model and introduces a database with non-uniform moisture distribution [7]. Compared with the XY model [7], the insulation information that can be reflected in this article is more abundant. Through the intersection frequency in Figure 17, reference data can be provided for the reverse verification of the evaluation result. However, in comparison with XY model, the evaluation method in this paper is more suitable for simpler insulating structures such as bushings with multilayer capacitive screens rather than transformers. Through FDS testing of different voltages and different wiring methods, the characteristic relaxation time constant-voltage curve can be drawn. The distribution position and non-uniformity coefficient *n* of water in the bushing can be obtained. Due to the introduction of variables such as voltage and wiring method, the macroscopic space charge polarization can be separated from multiple composite polarization processes. Compared with the results in reference [9], the relaxation time constant obtained by the experimental method in this paper is less affected.

## 5. Conclusions

In this paper, dielectric response tests in frequency domain are carried out for different types of dampness for transformer bushing. The mechanism of the influence of interface polarization on the dielectric response of the bushing is analyzed. FDS is used to diagnose the average moisture content and the degree of non-uniformity of bushing. In order to further clarify the radial moisture distribution, this paper analyzes the essential reason of the difference of the mobility of positive and negative ions in the damped oil paper and its influence on the ion concentration accumulation. On this basis, the contrast experiment and mechanism analysis of polarity reversal FDS are carried out for different bushing models with radial dampness. The macroscopic space charge polarization is separated from various polarization processes by changing the voltage and test wiring, and the curve characteristics, relaxation time constant, and evaluation method for evaluating the position of radial damp are formed. The specific contents are as follows:

(1)As the non-uniformity coefficient increases from 0 to 2, the tan*δ*-*f* curve of the non-uniform wetted sample fluctuates around the curve of the uniform sample, and shows a loss peak in the high frequency band. For the sample with average moisture content of 2%, 3%, and 4%, the combinations of 0.41% + 3.91%, 0.41% + 6.11%, and 2.03% + 6.11% have the largest fluctuation. There is a common intersection point between the tan*δ*-*f* curves of the non-uniform moisture samples with the same average moisture content. For the sample combinations with average water content of 2%, 3%, and 4%, the frequencies corresponding to their intersections are 0.0046 Hz, 0.15 Hz, and 1 Hz, respectively.(2)In the single-cycle FDS test of bushing model, when the positive electrode is first added to the area with higher moisture content, the amplitude of the tan*δ*-*f* curve is smaller. The tan*δ*-*f* curves under different wiring methods constitute a “ring-shaped” loss peak. As the voltage increases from 50V to 800V, the frequency corresponding to the loss peak of the tan*δ*-*f* curve difference decreases from 0.215 Hz to 0.046 Hz at 50 °C. As the temperature decreases from 50 ° C to 30 ° C, the frequency corresponding to the loss peak of the tan*δ*-*f* curve difference decreases from 0.046 Hz to below 0.001 Hz at a voltage of 800 V.

Therefore, by introducing the non-uniform damping database, the damping degree and damping type of the capacitive bushing can be evaluated. The frequency corresponding to the intersection point can be used as the limiting condition for multiple optimal solutions in database comparison. Further, by changing the wiring method and field strength, the parameters related to macroscopic space charge polarization and uneven moisture are separated from various polarization processes. The relationship between the time constant of space charge relaxation polarization and the insulation state can be established.

## Figures and Tables

**Figure 1 polymers-12-01219-f001:**
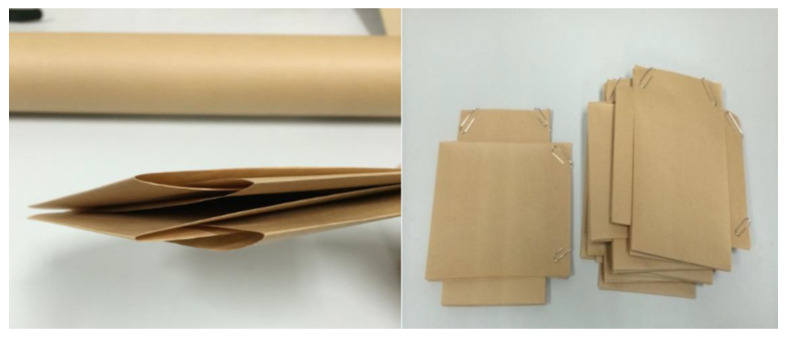
Overlapping cellulose polymer laminates.

**Figure 2 polymers-12-01219-f002:**
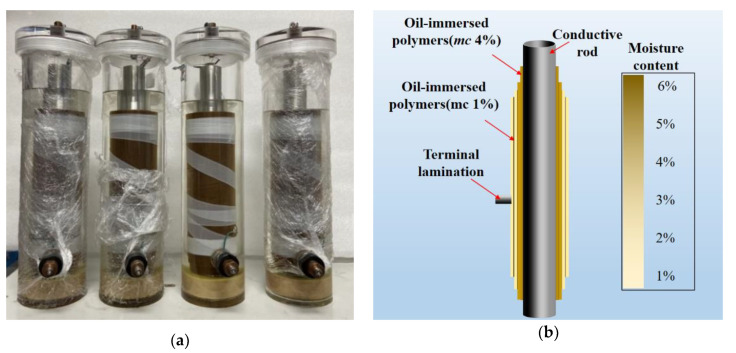
Bushing models diagram and moisture distribution diagram of 1#. (**a**) Bushing models with different types of moisture, (**b**) water distribution diagram of 1#.

**Figure 3 polymers-12-01219-f003:**
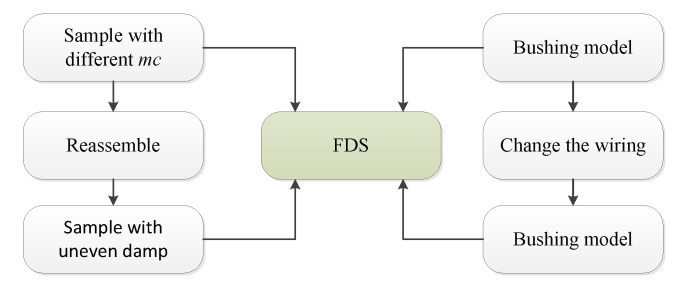
Experimental procedure and sample preparation.

**Figure 4 polymers-12-01219-f004:**
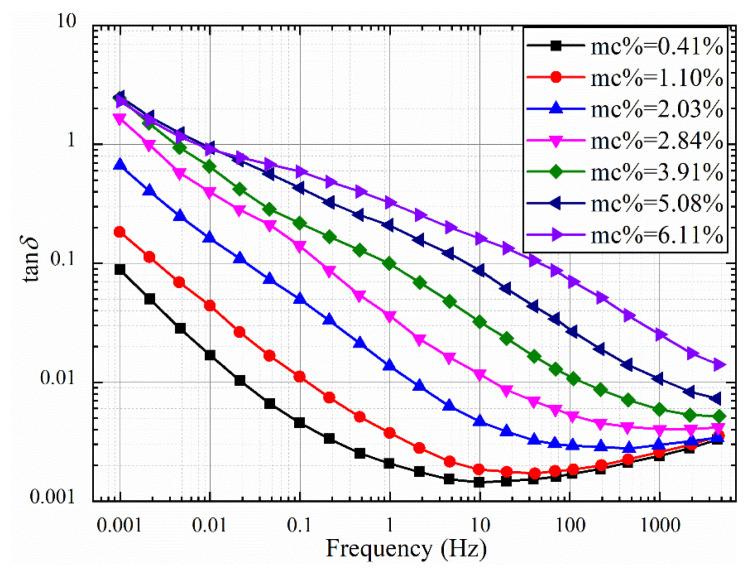
tan*δ*-*f* curves with different moisture contents.

**Figure 5 polymers-12-01219-f005:**
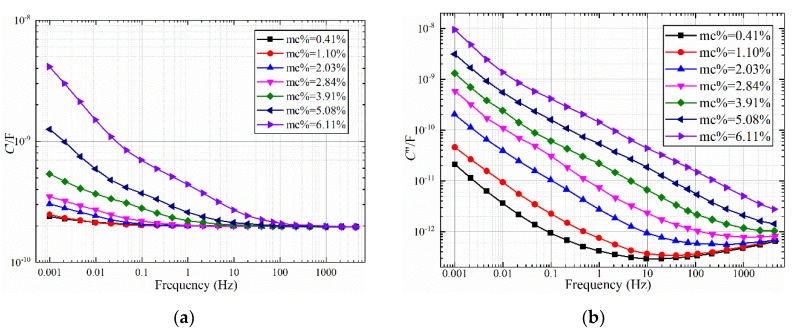
*C**-*f* curves with different moisture contents. (**a**) *C*′-*f*, (**b**) *C*″-*f*.

**Figure 6 polymers-12-01219-f006:**
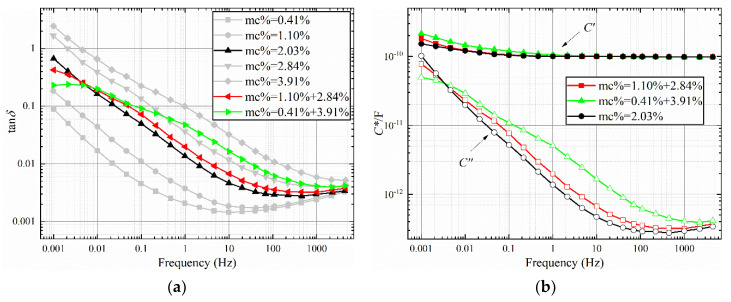
Frequency Domain Spectroscopy (FDS) characteristic curve with an average water content of 2%. (**a**) tan*δ*-*f*, (**b**) *C**-*f*.

**Figure 7 polymers-12-01219-f007:**
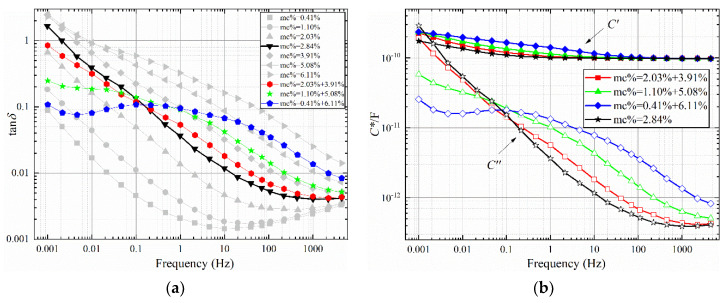
FDS characteristic curve with an average water content of 3%. (**a**) tan*δ*-*f*, (**b**) *C**-*f*.

**Figure 8 polymers-12-01219-f008:**
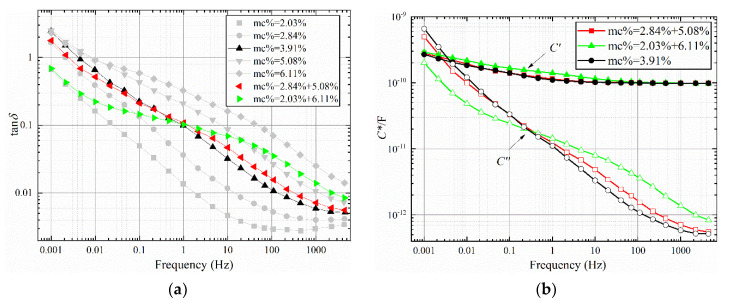
FDS characteristic curve with an average water content of 4%. (**a**) tan*δ*-*f*, (**b**) *C**-*f*.

**Figure 9 polymers-12-01219-f009:**
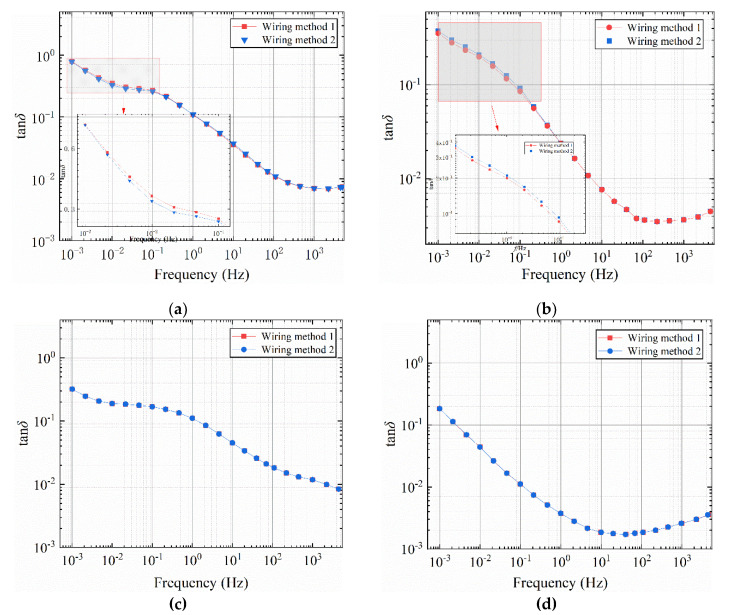
Effect of different wiring methods on tan*δ*-*f* curve of bushing model. (**a**) 1 # bushing model, (**b**) 2 # bushing model, (**c**) 3 # bushing model, (**d**) 4 # bushing model.

**Figure 10 polymers-12-01219-f010:**
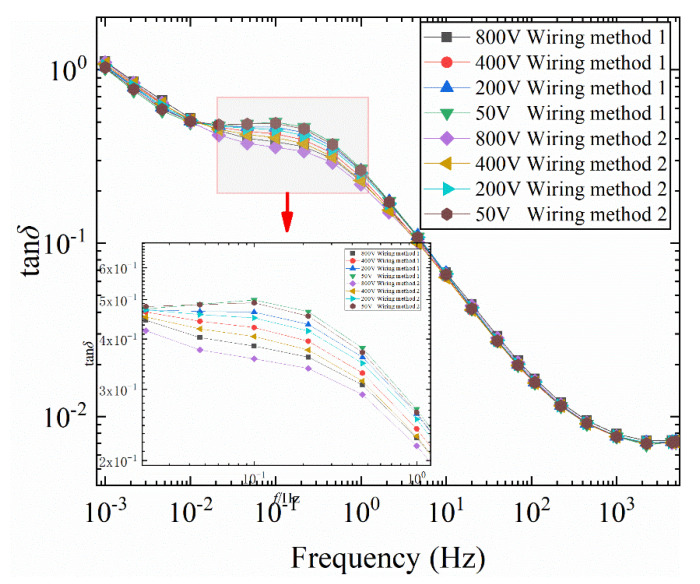
Influence of different wiring methods on tan*δ*-*f* curve of bushing model at different voltages (50 °C).

**Figure 11 polymers-12-01219-f011:**
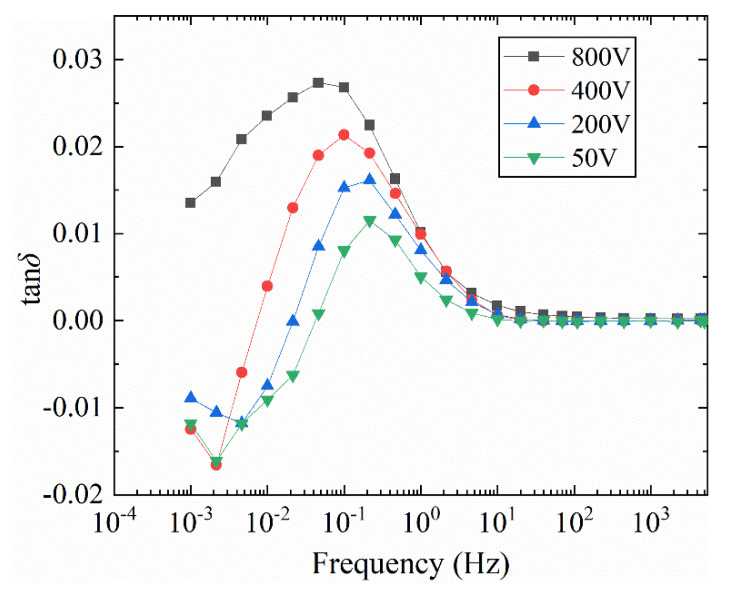
Influence of different wiring methods on the difference of tan*δ*-*f* curve of bushing model at different voltages (50 °C).

**Figure 12 polymers-12-01219-f012:**
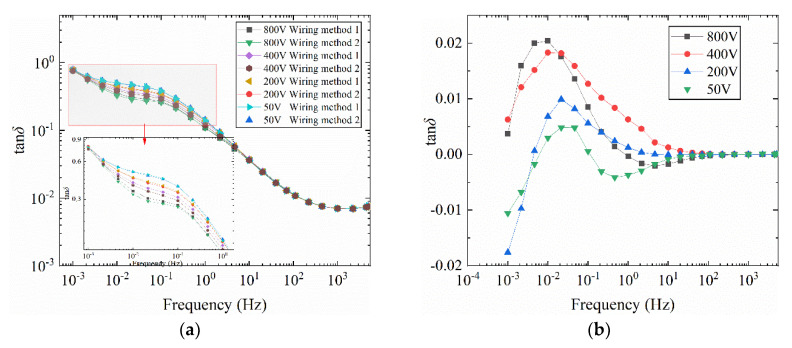
Influence of different wiring methods on tan*δ*-*f* curve of bushing model at different voltages (40 °C). (**a**) tan*δ*-*f*, (**b**) difference curves *of* tan*δ*-*f*.

**Figure 13 polymers-12-01219-f013:**
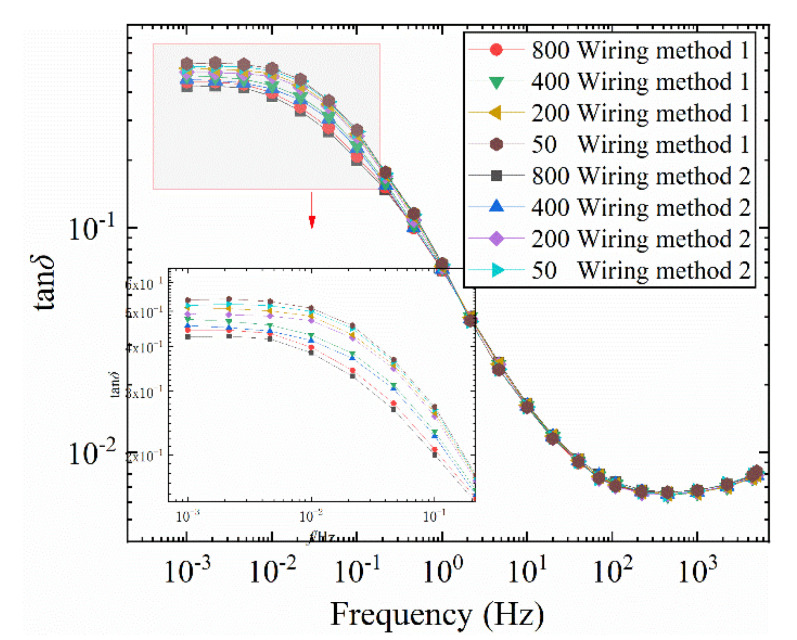
Influence of different wiring methods on tan*δ*-*f* curve of bushing model at different voltages (30 °C).

**Figure 14 polymers-12-01219-f014:**
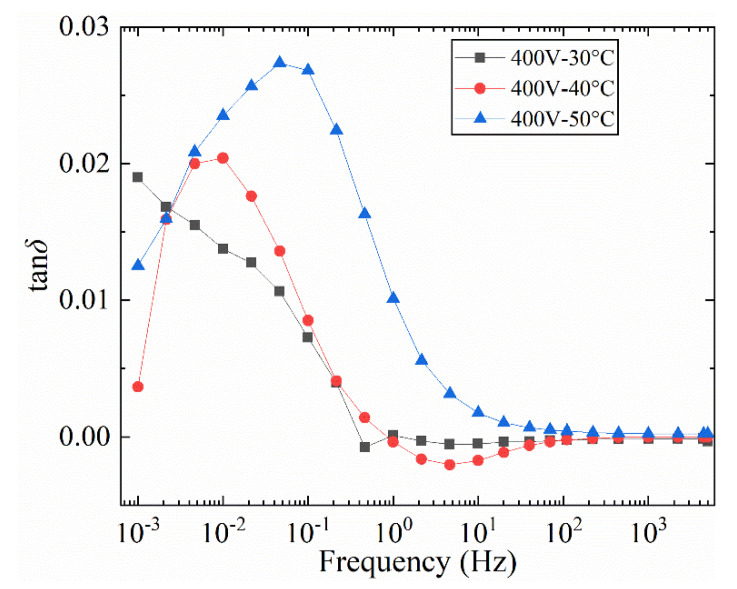
Influence of different temperatures on the difference of tan*δ*-*f* curve at the same voltage.

**Figure 15 polymers-12-01219-f015:**
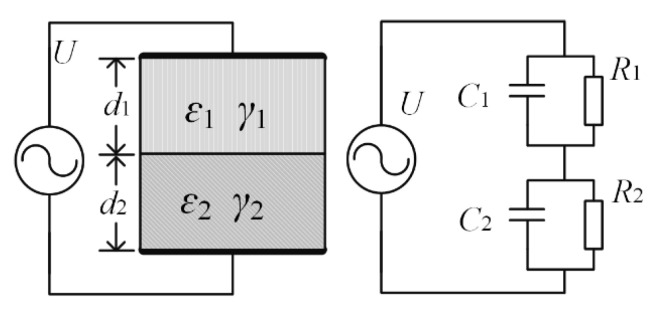
Equivalent circuit diagram of double-layer oil-paper insulation.

**Figure 16 polymers-12-01219-f016:**
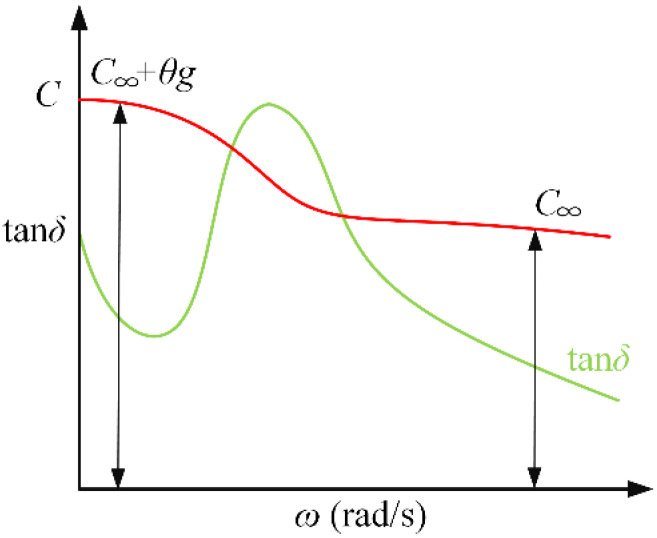
Relationship between tan*δ* and *ω*.

**Figure 17 polymers-12-01219-f017:**
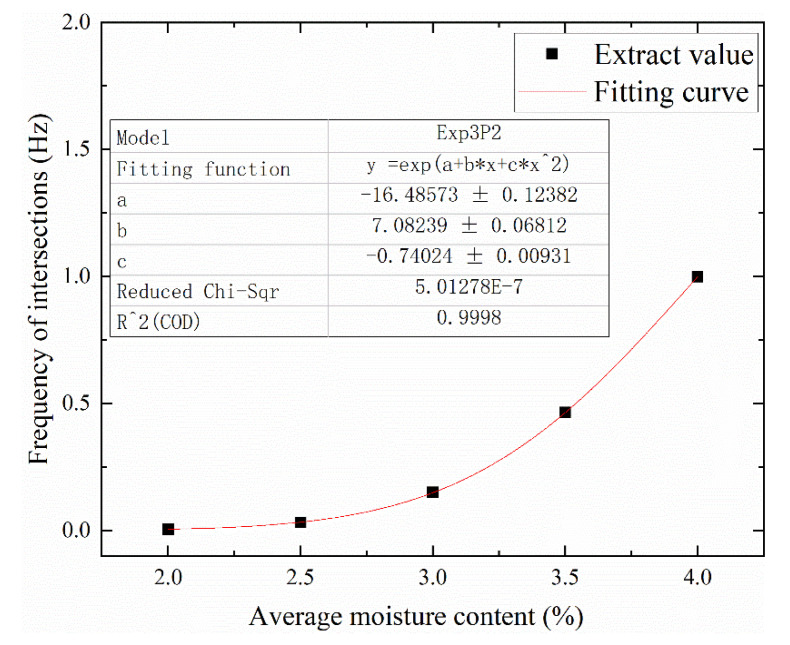
Relationship between average moisture content and frequency of intersection of tan*δ*-*f* curve (40 °C).

**Figure 18 polymers-12-01219-f018:**
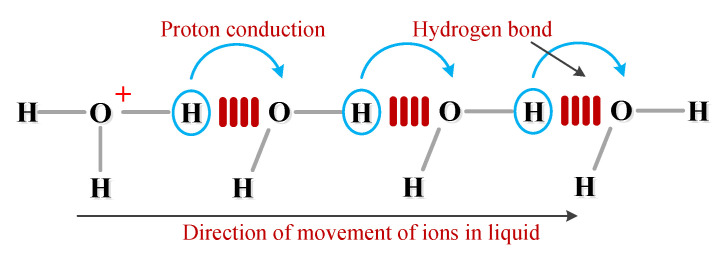
Proton conduction diagram of H_3_O^+^.

**Figure 19 polymers-12-01219-f019:**
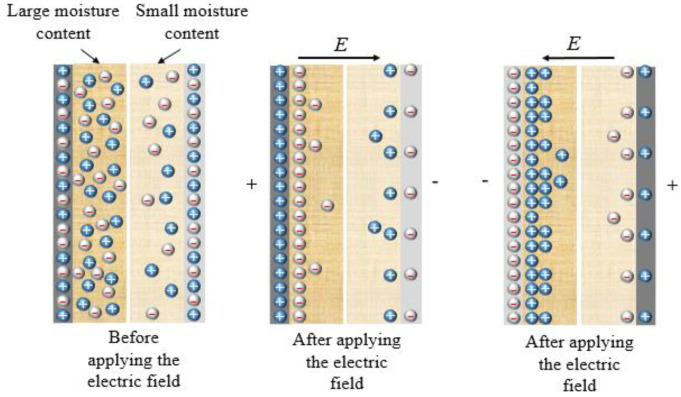
Ion distribution when moisture is non-uniform under AC electric field.

**Figure 20 polymers-12-01219-f020:**
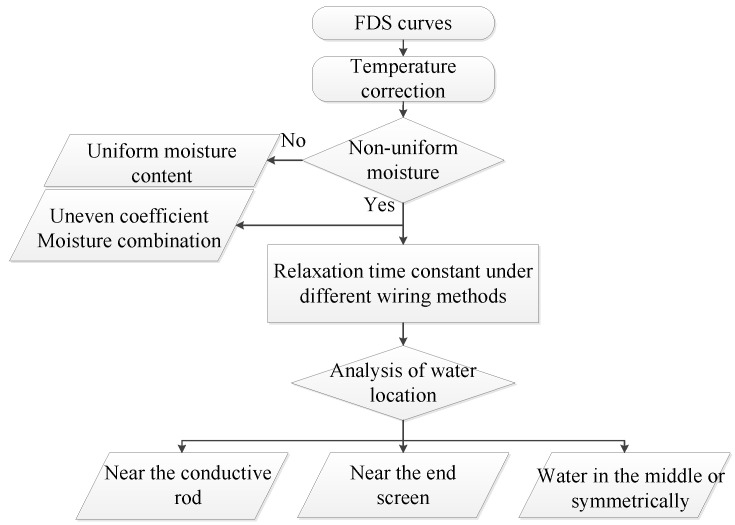
Flow chart for evaluating the distribution of moisture content.

**Table 1 polymers-12-01219-t001:** 26 kV Capacitive Bushing Core Size.

Plate Layer	Thickness (mm)	Upper Plate Difference (mm)	Lower Plate Difference (mm)	Plate Length (mm)
0				260
1	1.6	24	6	220
2	1.6	29	8	180
3	1.6	34	8	140
4	1.6	57	16	65

**Table 2 polymers-12-01219-t002:** Radial moisture distribution of oil-polymer insulated bushing model.

Serial Number	Laminate 1 mc (%)	Laminate 1 mc (%)	Laminate 1 mc (%)	Laminate 1 mc (%)	Damp Type
1#	4	4	1	1	inner
2#	1	1	4	4	outer
3#	1	4	4	1	middle
4#	1	1	1	1	uniform

**Table 3 polymers-12-01219-t003:** Combination of double oil-polymer samples with the same average moisture content.

Average Moisture Content (%)	Combination	*n*
2%	2% + 2%	0
	1% + 3%	1
	dry + 4%	2
3#	3% + 3%	0
	2% + 4%	2/3
	dry + 6%	4/3
4#	4% + 4%	0
	3% + 5%	1/2
	2% + 6%	1

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
