# Peer review of "Effect of Radial Moisture Distribution on Frequency Domain Dielectric Response of Oil-Polymer Insulation Bushing"

_polymers, 2020, doi:10.3390/polym12061219_

Round 1
Reviewer 1 Report
When I began to read the paper describing the « Effect of radial Moisture Distribution…” I was happy to continue my reading. It is clear that the literature is full of similar publications.
Your paper brings a clear view of the interaction of water with Oil-polymer, and I was happy to see that your results confirm the influence of the voltage and the frequency on the Voltage Breakdown Strength.
However, there are some corrections to be done. All these have to be addressed before published it.
1) Abstract: for the reader, please avoid “FDS” or similar symbol in your manuscript without writing the significance. The abstract should only talk/discuss the results and findings here.
2) Some sentences are to be corrected. Example: “Based on the problems ignored in the existing research, this paper proposes a method of radial dampness assessment based on interface polarization and macroscopic space charge polarization in FDS. The red part has to be revised or changed. The problems ignored etc.. are not ignored in other countries.
3) Some Figures are not easily readable. It is important that the reader can see easily the results inside. Figure 2b, figures 4, 5, 6, 7, 8, 9, until 14 for abscissa legends are to be modified. The Figures 9a, 9b, 9c, 9d, has to be placed on the same page!!!, Figures 16 and 17 are not readable too.
I would wish to see an evaluation the induced stress coming from the electric fields interaction
(dipoles and field apllied).

Reviewer 2 Report
The analyzed article is quite interesting and raises an important issue in the area of diagnostics of devices with high voltage insulation, namely the problem of moisture content assessment in polymer bushings.
The authors of the article presented in detail the motivation to conduct research. They also described in detail the method of sample preparation and the research methodology. The results of the research and their discussion are also presented clearly in the paper. Hence, I don't have a lot of comments for the article assessed. Some minor comments / questions are presented below:
- lines 38 and 39 - FDS and PDC appeared two times,
- line 70 - standards or standard?
- lines 84-86 - where do the data about field strength come from?
- line 177 - the authors wrote that "To avoid the impact between multiple tests, the interval between multiple tests is 10 minutes" why solely 10 minutes was chosen. Please explain it to the readers.
- line 199 - why the influence of voltage was considered in the paper? Please explain. How is the voltage level during commercial tests of transformers (bushings) based on FDS method?
- Eq 2 - please define Ip and Ic,
- Eq 7 - who is the author of this Eq? I mean how this Eq was developed?,
- could the authors add some comments on possible practical/engineering application of the results obtained within research conducted?
Reviewer 3 Report
- Introduction: From my point of view, introduction is not well focused. In a research paper, it is expected that introduction section briefly explains the starting background and, even more important, the originality (novelty) and relevancy of the study is well established. Once this is done, hypothesis and objectives of the study need to be addressed, as well as a brief justification of the conducted methodology. It is my belief that, in this case, authors do not put effort enough (or any effort) in highlighting the relevancy and (specially) the novelty of the study. Consequently, both major aspects are compromised. I strongly recommend that authors clearly explain all these aspects (including hypothesis and objectives) in order to add scientific rigor to the manuscript.
- Discussion: It is my opinion that a separate discussion section would help the reader to understand the study. However, the main issue arises from the lack of comparison with state-of-the-art studies. From my point of view, even though some publications are listed in the literature review, it has no justification not conducting a comparison with main previous works already published. The lack of such a comparison compromises the significance of the paper, so I strongly recommend authors to conduct as much and suitable comparisons as needed to solve this issue.
-
The conclusion is interestingly presented in bullet points. However, the obtained results should be briefly and clearly mentioned through the support of numerical data. What were the most sounding quantifiable findings of this study? I can see that the authors have a discussion section. However, that can’t be a conclusion. Otherwise, why is there a table in the middle of the conclusion? Thus, you can leave it as a discussion section and then add a conclusion.
- Figure 1 is very dubious. What is the purpose of this figure? Also, the presentation and quality of this figure is subpar.
